# Dynamic epigenetic age mosaicism in the human atherosclerotic artery

**Silvio Zaina** [1] *, **Manel Esteller** [2,3,4,5], **Isabel Gonçalves** [6], **Gertrud Lund** [7]

**1** Division of Health Sciences, Department of Medical Sciences, Leon Campus, University of Guanajuato, Leon, Mexico, **2** Josep Carreras Leukemia Research Institute, Badalona, Barcelona, Catalonia, Spain, **3** Centro de Investigación Biomédica en Red Cancer (CIBERONC), Madrid, Spain, **4** Institució Catalana de Recerca i Estudis Avançats (ICREA), Barcelona, Catalonia, Spain, **5** Physiological Sciences Department, School of Medicine and Health Sciences, University of Barcelona (UB), Barcelona, Catalonia, Spain, **6** Skåne University Hospital, Clinical Sciences Malmö, Lund University, Malmö, Sweden, **7** Department of Genetic Engineering, CINVESTAV Irapuato Unit, Irapuato, Mexico

* szaina@ugto.mx

**Data Availability Statement:** All relevant data are within the manuscript, its Supporting Information files and the URL indicated in Materials and Methods.

## Abstract

Accelerated epigenetic ageing, a promising marker of disease risk, has been detected in peripheral blood cells of atherosclerotic patients, but evidence in the vascular wall is lacking. Understanding the trends of epigenetic ageing in the atheroma may provide insights into mechanisms of atherogenesis or identify targets for molecular therapy. We surveyed DNA methylation age in two human artery samples: a set of donor-matched, paired atherosclerotic and healthy aortic portions, and a set of carotid artery atheromas. The well-characterized pan-tissue Horvath epigenetic clock was used, together with the Weidner whole-blood-specific clock as validation. For the first time, we document dynamic DNA methylation age mosaicism of the vascular wall that is atherosclerosis-related, switches from acceleration to deceleration with chronological ageing, and is consistent in human aorta and carotid atheroma. At CpG level, the Horvath epigenetic clock showed modest differential methylation between atherosclerotic and healthy aortic portions, weak association with atheroma histological grade and no clear evidence for participation in atherosclerosis-related cellular pathways. Our data suggest caution when assigning a unidirectional DNA methylation age change to the atherosclerotic arterial wall. Also, the results support previous conclusions that epigenetic ageing reflects non-disease-specific cellular alterations.

## Introduction

The concept that ageing is associated with causal changes in DNA, whether genetic or epigenetic, is at least half-a-century old [1]. A consequential hypothesis is that epigenetic and chronological age diverge in disease, thus providing a possible marker of disease risk. Microarray platforms that determine DNA methylation at CpG level with high reproducibility have provided opportunities to test that hypothesis in a variety of human conditions. In general, epigenetic clocks calculate a predicted age as weighted average of a core set of CpG that are selected based on their robust correlation with chronological age of healthy blood or tissue sample.

**Funding:** Supported by the following grants to ME: European Research Council (ERC) grant EPINORC under agreement no. 268626, the MICINN Project: SAF2011-22803, the Cellex Foundation, the Botin Foundation, the European Community's Seventh Framework Programme (FP7/2007–2013) from grant HEALTH-F5-2011–282510–BLUEPRINT and the Health and Science Departments of the Generalitat de Catalunya. SZ was supported by a CONACyT (Mexico) Sabbatical Fellowship no. 166058. ME is an Institució Catalana de Recerca i Estudis Avançats (ICREA) Research Professor. There was no additional external funding received for this study.

**Competing interests:** The authors have declared that no competing interests exist.

Epigenetic age acceleration or deceleration in a diseased tissue is determined by calculating the differential between predicted and chronological age [2]. Notably, epigenetic clock predicted age with higher precision than telomere length in one study [3]. Accelerated epigenetic ageing has been detected in peripheral blood cells of atherosclerotic patients, but evidence in the vascular wall is limited even outside the atherosclerosis field [4–7]. To our knowledge, epigenetic ageing of the vascular wall was addressed by two studies, one that tested for associations with haemostatic factors, and another that documented differential DNA methylation age (DNAmAge) between arteries and veins [8, 9]. In the present study, we use a within-donor comparison design to contrast DNAmAge of atherosclerotic and adjacent normal vascular tissue. Strengths of that design include: inter-individual confounders (genetic, environmental, stochastic variation) are reduced; it detects any intra-vascular bed mosaicism, a strong indication that epigenetic ageing is functionally associated with atherosclerosis and not just an epiphenomenon; it allows to dissect the role of systemic or local epigenetic ageing in atherogenesis. For those reasons, the description of epigenetic ageing in the atheroma may be relevant, as it potentially provides insights into mechanisms of aberrant gene expression that predispose to or accompany the natural history of atherosclerosis, and may identify potential epigenomic targets for molecular therapy. We surveyed DNAmAge of two sets of human atherosclerotic and healthy arteries for which we previously obtained extensive DNA methylome data, by using the well-characterized, pan-tissue Horvath clock and Weidner whole-blood-specific clock [3, 10, 11]. The Horvath clock uses 353 CpG as core age predictor and has been shown to accurately calculate the age of a variety of healthy human tissues, and to detect DNAmAge deviations from chronological age (*i.e.* acceleration or deceleration) in a range of diseases. Genes that harbour the 353 Horvath CpG are enriched in cell growth, cell survival and regulation of development functions. The Weidner clock has been designed to compute epigenetic age of blood cells. The clock uses three CpG as core predictor, located in genes that encode proteins involved in cell-cell communication (integrin, alpha 2b), aminoacid metabolism (aspartoacylase) and cAMP hydrolysis (phosphodiesterase 4C). The Weidner clock is expected to loosely match the Horvath clock, but we included both in our analysis reasoning that an identical direction of change of DNAmAge in the two clocks would be interpreted as validation of Horvath clock predictions.

## Materials and methods

Donor-matched pairs of atherosclerotic and histologically normal portions of the aorta (A-aorta and N-aorta, respectively; n = 15 pairs) obtained *post mortem*, carotid atheromas obtained by endarterectomy from symptomatic and asymptomatic patients (A-car; n = 19 each) and corresponding Infinium HumanMethylation450 BeadChip (Illumina) data were previously described [10, 12]. The characteristics of participating patients are reported in **S1 Table**. Aortic and carotid artery samples were obtained in two distinct environments (Spain and Sweden, respectively). The observation that 98% of differentially methylated CpG coincided between aortic and carotid artery samples suggests that batch or geographic origin effects were minimal [10]. Similarly, any significant effect of *post mortem* time to collection was ruled out in the original study, strongly supporting previous conclusions that DNA methylation profiles are stable after death and can be extrapolated to *ex vivo* samples [10, 13]. DNA methylation array data were extensively validated in the original study [10]. The study conformed to the principles of the Declaration of Helsinki and patients or relatives in the case of *post mortem* samples gave written consent prior to their participation. The protocol was approved by the Bellvitge Hospital ethical committee (authorization no. PR311/11). DNA methylation microarray data are freely available at the Gene Expression Omnibus database (www.ncbi.nlm.nih.

gov/geo/) with accession number GSE46401 [10]. To determine DNAmAge, Infinium HumanMethylation450 BeadChip data were formatted according to the calculator requirements and uploaded to the freely available epigenetic age calculator maintained by Horvath's laboratory at https://dnamage.genetics.ucla.edu/. The calculator yields predicted ages calculated with Horvath or Weidner clock algorithms and adjusts for sex and cell type. Output files generated by the calculator have been deposited on the University of Guanajuato cloud and are available at the native (non-shortened) URL: https://ugtomx-my.sharepoint.com/:f:/g/personal/szaina_ugto_mx/Eg-ccQD7c9dHjNekBqjRU1kBkg3zA895OzgJJWF24VZ1NA?e=W1l17j. T-test and one-way ANOVA were used for two group and multiple group comparisons. Pearson's r was used for correlation tests. Paired t-test of M-values was used to compare CpG methylation data [14]. Genome-wide significance was set at Bonferroni-corrected $p < 10^{-7}$ [10]. Statistical tests were performed with StatPlus Pro (AnalystSoft Inc.). Functional enrichment analysis of CpG-harbouring genes was performed with the DAVID tool (david.ncifcrf.gov) [15, 16].

## Results and discussion

Comparison of donor-matched aortas revealed a significant DNAmAge acceleration in A-aorta relative to N-aorta (average +4.3 years, $p = 8.0 \times 10^{-4}$) (**Fig 1A**). That differential DNAmAge was above the performance-assessed Horvath clock accuracy (3.6 years) and ~2 years higher than reported in peripheral blood of recurrent stroke patients [11, 17]. Weidner clock also yielded A-aorta DNAmAge acceleration although ~4.5-fold higher (+16.6 years, $p = 9.5 \times 10^{-5}$). These discrepancies are in line with previous comparisons of performance among clocks [18]. A-aorta and A-aorta-to-N-aorta differential DNAmAge did not differ between histological grades, except borderline significance in Weidner clock (Horvath: $p = 0.147$ and $p = 0.783$, respectively; Weidner: $p = 0.056$ and $p = 0.253$, respectively). Unexpectedly, N-aortas were on average younger relative to chronological age (Horvath: -4.9 years, $p = 0.018$; Weidner: -39.7 years, $p = 1.69 \times 10^{-9}$). Conversely, A-aortas DNAmAge matched chronological age (Horvath: -0.4 years, $p = 0.804$; Weidner: -1.1 years, $p = 0.421$). Also, the differential between either A-aorta or N-aorta DNAmAge and chronological age was strongly and inversely correlated with chronological age according to Horvath clock ($r = -0.72$ and $r = -0.89$, respectively, $p < 0.001$), while marginally significant in A-aortas by Weidner clock ($r = -0.48$, $p = 0.070$, and $r = -0.93$, $p = 10^{-6}$). That trend reflected the tendency for accelerated DNAmAge in chronologically younger aortas, followed by a deceleration at older chronological age (**Fig 1A**). This coincided with slow DNA methylation ageing across chronological age, as indicated by DNAmAge (y axis)/chronological age (x axis) slope <1 in N-aortas and A-aortas (0.46 and 0.57, respectively, in either Horvath or Weidner). Partially echoing our findings, peripheral blood DNAmAge is accelerated in young stroke patients but realigns with chronological age in older patients [4].

We replicated the above analysis in the A-car set. A-car and A-aorta were comparable: considering age-matched data points only (n = 14), A-car DNAmAge did not differ from A-aorta DNAmAge or from chronological age (Horvath: +2.4 years, $p = 0.167$ and +1.0 years, $p = 0.661$; Weidner: +2.5 years, $p = 0.124$ and +1.8 years, $p = 0.110$). The data confirms the high correlation previously observed between the two DNA methylation array data sets [10]. Conversely, A-car DNAmAge was accelerated relative to N-aorta (Horvath: +7.4 years, $p = 2.8 \times 10^{-4}$; Weidner: +24.7 years, $p = 3.1 \times 10^{-16}$). Mirroring the aorta data set, the differential between A-car DNAmAge and chronological age was inversely correlated with chronological age (Horvath: $r = -0.75$, $p < 10^{-6}$; Weidner: $r = -0.56$, $p = 2.6 \times 10^{-4}$). Also, a tendency to DNAmAge acceleration followed by deceleration with chronological age was observed in A-car,

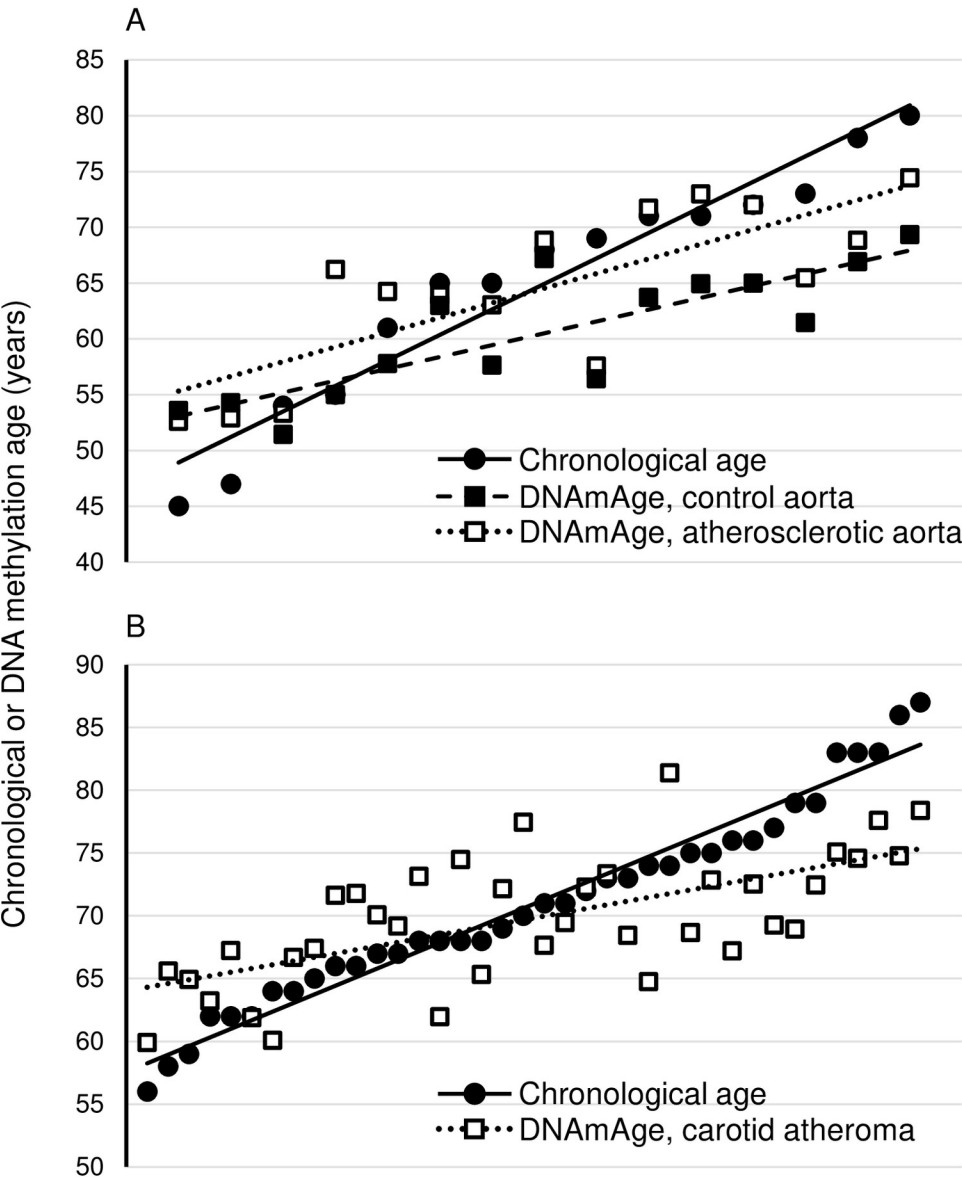

**Fig 1. Chronological and Horvath DNA methylation age in aortas and carotid atheromas.** Samples are ordered by increasing chronological age (reported on the y-axis) and positioned at regular intervals on the x-dimension. A, donor-matched atherosclerotic and histologically normal portions of the aorta; B, carotid atheromas. Interpolating lines are not regression slopes, rather are intended to convey differences in ageing pace. Notice the tendency for accelerated and decelerated DNA methylation age at early and later ages, respectively.

concomitant with a slower DNA methylation ageing than chronological ageing (DNAmAge (y axis)/chronological age (x axis) slope <1: Horvath: 0.44; Weidner: 0.22) (**Fig 1B**). Furthermore, the A-car data set allowed to explore associations of DNAmAge with asymptomatic/symptomatic status and with symptom-to-endarterectomy time. DNAmAge was not different between symptomatic and asymptomatic patient atheromas (Horvath: p = 0.116; Weidner: p = 0.151). Likewise, no significant association was detected of DNAmAge with symptom-to-endarterectomy time (Horvath: r = -0.104; Weidner: r = 0.173; p>0.307 in either clock).

**Table 1. Horvath epigenetic clock CpG (n = 353) that are significantly different between atherosclerotic and healthy portions of the aorta.**

| CpG ID | A-N[¶] Δβ | p[§] | Chromosome: position[‡] | Gene symbol | Relevance for atherosclerosis |
|---|---|---|---|---|---|
| cg01353448 | 0.13 | $6.23 \times 10^{-8}$ | 2:74,609,742 | HTRA2 | Unknown |
| cg01656216 | 0.16 | $4.11 \times 10^{-8}$ | 9:85,761,834 | C9orf64 | Unknown |
| cg14329157 | 0.05 | $1.80 \times 10^{-9}$ | 16:23,597,303 | PLK1 | Overexpressed in atherosclerosis [19] |
| cg22809047 | 0.08 | $8.97 \times 10^{-8}$ | 17:71,487,684 | ACOX1 | Involved in inflammation resolution [20] |

[¶]A and N, atherosclerotic and healthy portion of the aorta, respectively.

[§]Comparison of M-values, paired t-test, Bonferroni-corrected (threshold: $p = 10^{-7}$).

[‡]Genome build 36.

In order to gain insights into possible mechanisms underlying the above observations, we assessed the methylation status of the 353 Horvath clock CpG in A-aorta and N-aorta samples [11]. Four CpG (or 1.13%) were differentially methylated between A-aorta and N-aorta at genome-wide significance ($p < 10^{-7}$, Bonferroni correction) (**Table 1**, **S1 Fig**). Since differences of methylation β were below 16% in all cases, those CpG did not pass differential methylation criteria in the original analysis of A-aorta and N-aorta [10]. Those four CpG map to proximal (cg01656216) or distal (cg01353448, cg14329157, cg22809047) promoters. Comparatively little or no published evidence links the four corresponding genes to atherosclerosis. HtrA serine peptidase 2 (HTRA2) and chromosome 9 open reading frame 64 (C9orf64) have not been implicated in atherogenesis. Limited evidence indicates that polo like kinase 1 (PLK1), encoding a stimulator of cell proliferation, and acyl-CoA oxidase 1 (ACOX1), encoding a player in fatty acid metabolism, are upregulated in peripheral blood cells of atherosclerotic patients and linked to resolution of inflammation, respectively [19, 20]. Furthermore, less than 10% of Horvath clock CpG were correlated with chronological age or histological grade (30 and 14, respectively) at nominal significance level (overall lowest significance $p = 4.7 \times 10^{-4}$), but none reached genome-wide significance ($p < 10^{-7}$;) (**S2 Table**, **S1 Fig**). None of those 44 CpG was differentially methylated between A-aorta and N-aorta. No significant functional enrichment (Gene_Ontology, Pathways, Protein_Interactions, Tissue_Expression) was observed for the corresponding genes in either case. The majority (~78.6%) of histological grade-associated Horvath clock CpG were also correlated with chronological age, reflecting the fact that chronological age tended to be lower in grade III atheromas than in grade VII counterparts ($p = 0.055$; ANOVA, Fisher LSD *post hoc*). Taken together, the analysis of Horvath clock CpG confirmed previous conclusions that epigenetic ageing reflects alterations in fundamental, rather than disease-specific, cellular epigenetic maintenance systems (the concept of EMS discussed in [11]). A few studies support that notion. Peripheral blood cell profiling of six cohorts spanning a wide range of chronological age and pathological conditions, revealed that a minority of Horvath clock CpG correlate with expression of genes *in cis* that are not significantly enriched in any Gene Ontology term [21]. Epigenetic ageing was weakly associated with classical cardiovascular risk factors in the large MESA and HRS cohorts [22]. Furthermore, no association of epigenetic age with development of cardiovascular risk factors was observed in a cohort of 6–10 years old children [23]. The results of the latter two studies agree with our observation that epigenetic ageing was not associated with asymptomatic/symptomatic status in the carotid atheroma. As *caveat*, Horvath clock CpG are associated with gene expression *in trans* in loci that regulate T cell function and may therefore be relevant for atherosclerosis [21, 24, 25]. As the atheroma discontinuously expands by bursts in macrophage infiltration and cellular proliferation, it is possible that the epigenetic clock is a readout of mitotic age in atherosclerosis [26]. Yet, that hypothesis runs counter to the fact that the Horvath epigenetic

clock was previously found to be independent of mitotic age and was not associated with histological grade—a proxy of the proliferative history of the atheroma—in this study. In the light of the mentioned evidence, epigenetic ageing may be the consequence of poorly understood cellular mechanisms [27], or may simply be a useful epiphenomenon for marker identification purposes.

In conclusion, we document for the first time dynamic epigenetic age mosaicism of the vascular wall that is atherosclerosis-related, switches from acceleration to deceleration with chronological ageing, and is consistent in human aorta and carotid atheromas. Our data suggest caution when assigning unidirectional DNAmAge change to the atherosclerotic artery. Additionally, Horvath clock CpG methylation profiles reinforce the notion that epigenetic ageing reflects poorly understood, non-disease-specific cellular events.

## Supporting information

**S1 Fig. Horvath clock CpG that were differentially methylated between atherosclerotic and healthy aortic portions, or associated with chronological age or atheroma histological grade.** Differential methylation was significant at genome-wide level ($p < 10^{-7}$, Bonferroni correction; see Table 1 for details). Other associations were significant at nominal significance level ($p < 0.05$; see S1 Table for details).
(PPTX)

**S1 Table. Patient information.**
(DOCX)

**S2 Table. CpG associated with chronological age or histological grade.**
(XLSX)

## Author Contributions

**Conceptualization:** Silvio Zaina, Gertrud Lund.

**Funding acquisition:** Manel Esteller.

**Writing – original draft:** Silvio Zaina.

**Writing – review & editing:** Manel Esteller, Isabel Gonçalves, Gertrud Lund.

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
