## [Decision Letter · Decision Letter 0]

27 Apr 2022

PONE-D-22-05348Dynamic epigenetic age mosaicism in the human atherosclerotic arteryPLOS ONE

Dear Dr. Zaina,

Thank you for submitting your manuscript to PLOS ONE. After careful consideration, we feel that it has merit but does not fully meet PLOS ONE’s publication criteria as it currently stands. Therefore, we invite you to submit a revised version of the manuscript that addresses the points raised during the review process.

The reviewers where splitted on the value of the manuscript, therefore you need to address in your revision all comments in great details.

We look forward to receiving your revised manuscript.

Kind regards,

Osman El-Maarri, Ph.D

Academic Editor

PLOS ONE

Journal Requirements:

(Please include your amended Funding Statement within your cover letter. We will change the online submission form on your behalf."

Supported by the following grants to ME: European Research Council (ERC) grant EPINORC under agreement no. 268626, the MICINN Project: SAF2011-22803, the Cellex Foundation, the Botin Foundation, the European Community’s Seventh Framework Programme (FP7/2007– 2013) from grant HEALTH-F5-2011–282510–BLUEPRINT and the Health and Science Departments of the Generalitat de Catalunya. SZ was supported by a CONACyT (Mexico) Sabbatical Fellowship no. 166058. ME is an Institució Catalana de Recerca i Estudis Avançats (ICREA) Research Professor.)

Reviewers' comments:

Reviewer's Responses to Questions

**Comments to the Author**

1. Is the manuscript technically sound, and do the data support the conclusions?

Reviewer #1: Yes

Reviewer #2: Yes

Reviewer #3: No

2. Has the statistical analysis been performed appropriately and rigorously? 

Reviewer #1: Yes

Reviewer #2: Yes

Reviewer #3: No

3. Have the authors made all data underlying the findings in their manuscript fully available?

Reviewer #1: Yes

Reviewer #2: Yes

Reviewer #3: Yes

4. Is the manuscript presented in an intelligible fashion and written in standard English?

Reviewer #1: Yes

Reviewer #2: Yes

Reviewer #3: Yes

5. Review Comments to the Author

Reviewer #1: Review for “Dynamic epigenetic age mosaicism in the human atherosclerotic artery”

Zaina and colleagues utilized 2 datasets to evaluate associations of epigenetic aging in atherosclerotic aortas, normal aortas, and carotid atheromas using the Illumina Infinium HumanMethylation450 BeadChip. Through their analyses, they found that atherosclerotic aortas had greater age acceleration when compared to normal aortas. However, in comparison to chronological age, atherosclerotic aortas showed no acceleration, and normal aortas showed a deceleration in aging. An analysis of carotid atheromas showed accelerated epigenetic aging when compared to normal aortas, but when compared to atherosclerotic aortas, there was no difference.

Methods and Materials:

Line 55: Were there any location/batch effects observed?

Line 58: Do you expect there to be differences between post mortem and live tissue samples? Would this impact the results?

Results:

1) Your graph shows a non-linear relationship for atherosclerotic heart. I’m not sure if you can make a comparison with chronological age. Fitting a spline might help.

Figures:

1) Is the x-axis meaningful? A legend mapping out the meaning of circles etc. would be helpful. Reading it in the figure legend at the bottom makes it difficult to read.

1A) Could this be split into 2 graphs? It’s difficult to read.

S1) Including the p-value thresholds in the figure legend will be helpful.

Reviewer #2: The manuscript under the title: “Dynamic epigenetic age mosaicism in the human atherosclerotic artery” represents an interesting original scientific paper describing the role of epigenetic aging (DNA methylation) in the settings of donor-matched, paired atherosclerotic, and healthy aortic portions and in the settings of carotid artery atheroma.

However, some obstacles prevent the manuscript from publication in its current form.

These, among others, include:

The Materials and method section should be more extensive. Although the characteristics of samples used in the current study were, according to the authors, described in the original research (Zaina et al., 2014), some basic patient information should be described in the current study as well.

The description of the pan-tissue Horvath epigenetic clock and the Weidner whole-blood-specific clock and the calculation of DNAmAge should be more detailed to be more understandable to the scientists outside the field.

The authors have written. “Those four CpG map to proximal (cg01656216) or distal promoters”. - The distal promoters should be, if possible, mentioned as well.

The authors have written: “Comparatively little, or no published evidence links the four corresponding genes to atherosclerosis.” - The corresponding four genes should be mentioned in the text.

It seems, at least to me, that most of the data described in the Result section are not illustrated in the form of a table or figure presentation.

A major revision of the manuscript is recommended.

Reviewer #3: The authors aimed to test epigenetic aging in human artery samples, a set of donor-matched,

paired atherosclerotic and healthy aortic portions, and a set of carotid artery atheromas. I do not get the point of the analysis since we all know that epigenetic modifications are tissue specific. Since a lot of epigenetic sites were available, it would be good to analyze these as the main part, which, however, has been published by this group. Thus, I may recommend rejection at this point as this work does not add new to the field. Also the results were not well organized and it seem hard to get what the authors wanted to show.

6. PLOS authors have the option to publish the peer review history of their article (what does this mean?). If published, this will include your full peer review and any attached files.

Reviewer #1: No

Reviewer #2: No

Reviewer #3: No

---

## [Author Response · Author response to Decision Letter 0]

2 May 2022

Dear Dr. El-Maarri,

We are grateful for the opportunity to submit a revised version of our manuscript PONE-D-22-05348 "Dynamic epigenetic age mosaicism in the human atherosclerotic artery" and appreciate your time and effort to find three independent Reviewers.

Please find in the next pages a detailed rebuttal to the points raised by the Reviewers. We responded to the third Reviewer's strong criticism by pitching the merit of the study the best way we could.

We also addressed the additional requirements that you indicated in the decision email as follows:

1. We checked the manuscript to adhere to PLOS ONE's style requirements.

2. We submit an amended funding statement that reads as follows:

Supported by the following grants to ME: European Research Council (ERC) grant EPINORC under agreement no. 268626, the MICINN Project: SAF2011-22803, the Cellex Foundation, the Botin Foundation, the European Community’s Seventh Framework Programme (FP7/2007–2013) from grant HEALTH-F5-2011–282510–BLUEPRINT and the Health and Science Departments of the Generalitat de Catalunya. SZ was supported by a CONACyT (Mexico) Sabbatical Fellowship no. 166058. ME is an Institució Catalana de Recerca i Estudis Avançats (ICREA) Research Professor. There was no additional external funding received for this study.

We hope that the amended funding statement is acceptable. As you indicated in the decision email, the statement is part of this cover letter but has not been changed in the manuscript.

3. We make a minimal data set - i.e. output files generated by the DNA methylation age calculator - available from the cloud of the University of Guanajuato, to which I am affiliated. We specify the corresponding link in the Methods section (lines 88-89).

Please let us know whether any required information is missing.

We look forward to the Editorial decision.

Sincerely,

Silvio Zaina

Reviewer #1: Review for “Dynamic epigenetic age mosaicism in the human atherosclerotic artery” Zaina and colleagues utilized 2 datasets to evaluate associations of epigenetic aging in atherosclerotic aortas, normal aortas, and carotid atheromas using the Illumina Infinium HumanMethylation450 BeadChip. Through their analyses, they found that atherosclerotic aortas had greater age acceleration when compared to normal aortas. However, in comparison to chronological age, atherosclerotic aortas showed no acceleration, and normal aortas showed a deceleration in aging. An analysis of carotid atheromas showed accelerated epigenetic aging when compared to normal aortas, but when compared to atherosclerotic aortas, there was no difference.

Methods and Materials:

Line 55: Were there any location/batch effects observed?

Authors' response: We addressed this important point in the original study (Methods, lines 72-74) and in the Results and Discussion section (lines 125-126).

Line 58: Do you expect there to be differences between post mortem and live tissue samples? Would this impact the results?

Authors' response: No correlation between post mortem time at specimen collection and methylation of relevant loci was observed in the original study (Zaina et al., page 696). This is strong evidence for DNA methylation stability across our specimen, irrespective of origin. We added this valid point to the Methods section (line 74-77).

Results:

1) Your graph shows a non-linear relationship for atherosclerotic heart. I’m not sure if you can make a comparison with chronological age. Fitting a spline might help.

Authors' response: Sorry for not conveying the substance of the data shown in Figure 1. The non-linearity is visually true but misleading, as it reflects a slightly wider distribution of chronological age at the two extremes of the age range. Please note that samples are equally spaced along the x-dimension. The x-axis does not represent any biological variable that describes the samples in any way. We do not want to dodge this valid comment, but fitting the curve would have a weak biological justification. We maintain that interpolating lines are in this case more illustrative than non-linear fitting.

Figures:

1) Is the x-axis meaningful? A legend mapping out the meaning of circles etc. would be helpful. Reading it in the figure legend at the bottom makes it difficult to read.

Authors' response: We modified Figure 1 accordingly, hoping that quality improved. The figure has no x-axis. Samples were ordered by chronological age (reported on the y-axis) and positioned at regular intervals on the x-dimension. This rationale in constructing the figure has been added to the legend. We hope that this adequately addresses the reviewer's concern.

1A) Could this be split into 2 graphs? It’s difficult to read.

Authors' response: We gave a lot of thought to this comment. The three data sets in Figure 1A are linked and splitting them into two panels would mean either plotting DNA methylation age alone without chronological age or plotting chronological age twice with DNA methylation age of control aorta in one panel, and of atherosclerotic aorta in the other. Again, we do not want to dodge the Reviewer's point; but we feel that by adopting either solution readability would not improve.

S1) Including the p-value thresholds in the figure legend will be helpful.

Authors' response: We changed the figure legend accordingly and have actually indicated the threshold p values in Table I (line 181) and in the text (line 92, line 152). Thanks for the suggestion.

 

Reviewer #2: The manuscript under the title: “Dynamic epigenetic age mosaicism in the human atherosclerotic artery” represents an interesting original scientific paper describing the role of epigenetic aging (DNA methylation) in the settings of donor-matched, paired atherosclerotic, and healthy aortic portions and in the settings of carotid artery atheroma. However, some obstacles prevent the manuscript from publication in its current form. These, among others, include:

The Materials and method section should be more extensive. Although the characteristics of samples used in the current study were, according to the authors, described in the original research (Zaina et al., 2014), some basic patient information should be described in the current study as well.

Authors' response: We thank this reviewer for the overall encouraging comments. We have added a supplemental table (S1 Table) with patient information (lines 70-71).

The description of the pan-tissue Horvath epigenetic clock and the Weidner whole-bloodspecific clock and the calculation of DNAmAge should be more detailed to be more understandable to the scientists outside the field.

Authors' response: The Reviewer is right. We rewrote and expanded the Introduction section (lines 31-65). It now contains an essential description of the broad rationale behind epigenetic clocks and of the loci used to predict age by Horvath or Weidner clocks. Also, we now outline the methodology used to calculate the predicted age (Methods section, lines 82-86) and made the output files generated by the calculator available (lines 86-89). In Results, we added a sentence on variation among epigenetic clocks (lines 100-101).

The authors have written. “Those four CpG map to proximal (cg01656216) or distal promoters”.

- The distal promoters should be, if possible, mentioned as well.

Authors' response: We changed that sentence accordingly (line 144).

The authors have written: “Comparatively little, or no published evidence links the four corresponding genes to atherosclerosis.” - The corresponding four genes should be mentioned in the text.

Authors' response: We mention the gene names and biological function and cite relevant literature in the text (lines 145-150).

It seems, at least to me, that most of the data described in the Result section are not illustrated in the form of a table or figure presentation.

Authors' response: We carefully revised the text for data to be included in new figures or tables. The first segment of the Results section (lines 96-115) describes data that are included in Figure 1A or statistical analysis of those data that if spelled out in the figure would compromise readability. The same can be said of the part of Results describing carotid artery samples and referring to Figure 1B (lines 122-137). As for the segment describing individual CpG (lines 138-177), we added a supplemental table (S2 Table; line 152-153) that describes CpG correlation statistics and genomic position, and the identity of genes that harbour the chronological age-associated or histological grade-associated Horvath CpG. In our view, the new S2 Table should provide a complete information about the results of the analysis at CpG level, together with and complementing S1 Figure. We hope that this Reviewer will now find that tables and figures sufficiently cover our results, taking into account that the new S1 Table was also added (see above).

A major revision of the manuscript is recommended.

Authors' response: We have striven to revise the manuscript the best way we could and hope that we appropriately addressed the Reviewer's concerns. 

Reviewer #3: The authors aimed to test epigenetic aging in human artery samples, a set of donor-matched, paired atherosclerotic and healthy aortic portions, and a set of carotid artery atheromas. I do not get the point of the analysis since we all know that epigenetic modifications are tissue specific. Since a lot of epigenetic sites were available, it would be good to analyze these as the main part, which, however, has been published by this group. Thus, I may recommend rejection at this point as this work does not add new to the field. Also the results were not well organized and it seem hard to get what the authors wanted to show.

Authors' response: We are sorry for the bad impression our manuscript left this Reviewer with. We have revised the manuscript based on the other two Reviewers' criticism. This Reviewer may thererfore find the new manuscript version more intelligible. It is true that DNA methylation profiles are in part tissue-specific. However, in this study we addressed a distinct question: whether epigenetic ageing is mosaic within the same vascular bed and particularly between atherosclerotic and healthy portions. We believe that this approach has been overlooked, as most epigenetic age studies address ageing in whole blood and use that information as proxy of the organism's age to compare epigenetic ageing pace among individuals. We tried to explain that epigenetic age mosaicism within an artery would have significant implications: divergence of ageing would provide strong indication that the epigenetic clock is functionally associated with atherogenesis, and that epigenetic ageing is local rather than systemic. Although that would not be enough per se to deduce causality, it may help to understand the molecular basis of atherosclerosis and identify therapeutic targets (Introduction section, lines 46-53). We respectfully ask this Reviewer to consider our rebuttal and to take a fresh look at our results.

---

## [Decision Letter · Decision Letter 1]

23 May 2022

Dynamic epigenetic age mosaicism in the human atherosclerotic artery

PONE-D-22-05348R1

Dear Dr. Zaina,

We’re pleased to inform you that your manuscript has been judged scientifically suitable for publication and will be formally accepted for publication once it meets all outstanding technical requirements.

Kind regards,

Osman El-Maarri, Ph.D

Academic Editor

PLOS ONE

Reviewers' comments:

Reviewer's Responses to Questions

**Comments to the Author**

1. If the authors have adequately addressed your comments raised in a previous round of review and you feel that this manuscript is now acceptable for publication, you may indicate that here to bypass the “Comments to the Author” section, enter your conflict of interest statement in the “Confidential to Editor” section, and submit your "Accept" recommendation.

Reviewer #1: All comments have been addressed

Reviewer #2: All comments have been addressed

2. Is the manuscript technically sound, and do the data support the conclusions?

Reviewer #1: Yes

Reviewer #2: Yes

3. Has the statistical analysis been performed appropriately and rigorously? 

Reviewer #1: Yes

Reviewer #2: Yes

4. Have the authors made all data underlying the findings in their manuscript fully available?

Reviewer #1: Yes

Reviewer #2: Yes

5. Is the manuscript presented in an intelligible fashion and written in standard English?

Reviewer #1: Yes

Reviewer #2: Yes

6. Review Comments to the Author

Reviewer #1: (No Response)

Reviewer #2: The authors have addressed all the questions and suggestions raised by the reviewers. The revised version of the manuscript is suitable for publication in its current form.

7. PLOS authors have the option to publish the peer review history of their article (what does this mean?). If published, this will include your full peer review and any attached files.

Reviewer #1: No

Reviewer #2: No

---

## [Editor Report · Acceptance letter]

27 May 2022

PONE-D-22-05348R1 

Dynamic epigenetic age mosaicism in the human atherosclerotic artery 

Dear Dr. Zaina:

I'm pleased to inform you that your manuscript has been deemed suitable for publication in PLOS ONE. Congratulations! Your manuscript is now with our production department. 

Kind regards, 

on behalf of

Priv.-Doz. Dr. Osman El-Maarri 

Academic Editor

PLOS ONE